# Efficient Process Reward Model Training via Active Learning

**Keyu Duan**[1,2*]  **Zichen Liu**[1,2]   **Xin Mao**[1]   **Tianyu Pang**[1]   **Changyu Chen**[1,3]

**Qiguang Chen**   **Michael Qizhe Shieh**[2†]  **Longxu Dou**[1†]

[1]Sea AI Lab    [2]National University of Singapore    [3]Singapore Management University

## Abstract

Process Reward Models (PRMs) provide step-level supervision to large language models (LLMs), but scaling up training data annotation remains challenging for both humans and LLMs. To address this limitation, we propose an active learning approach, ACTPRM, which proactively selects the most uncertain samples for training, substantially reducing labeling costs. During training, we use the PRM to estimate uncertainty after the forward pass, retaining only highly uncertain data. A capable yet costly reasoning model then labels this data. Then we compute the loss w.r.t. the labels and update the PRM's weights. We compare ACTPRM vs. vanilla fine-tuning, on a pool-based active learning setting, demonstrating that ACTPRM reduce 50% annotation, but achieving the comparable or even better performance. Beyond annotation efficiency, we further advance the actively trained PRM by filtering over 1M+ math reasoning trajectories with ACTPRM, retaining 60% of the data. A subsequent training on this selected dataset yields a new state-of-the-art (SOTA) PRM on ProcessBench (75.0%) and PRMBench (65.5%) compared with same sized models [1].

## 1 Introduction

Recently, Large Language Models (LLMs) (DeepSeek-AI et al., 2025; Yang et al., 2024; OpenAI et al., 2024b) have shown remarkable advances in mathematical reasoning, yet they can make mistakes during chain-of-thought (CoT) reasoning despite correct final answers(Zheng et al., 2024). To address this challenge, process reward models (Lightman et al., 2023; Wang et al., 2024; Zhang et al., 2025) were proposed, aiming to identify process errors and provide finer-grained supervision of the training process.

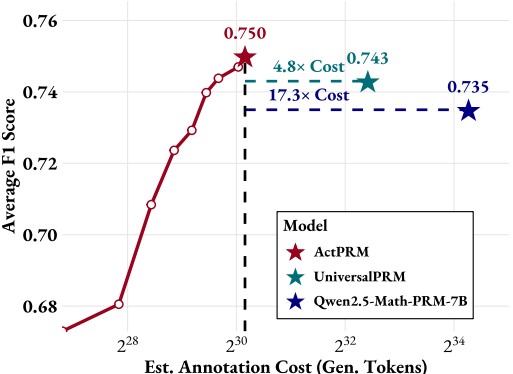

Figure 1: Average F1 score on Process-Bench (Zheng et al., 2024) versus estimated annotation cost. ACTPRM outperforms prior SOTA models while requiring merely 20% of the annotation costs.

The main challenge in training Process Reward Models (PRMs) lies in obtaining fine-grained step-level annotations, which remain prohibitively expensive. Lightman et al. (2023) pioneered PRM training by employing human experts to label 75K questions at the step level. While their approach achieved high-quality results (reaching 57.5% on ProcessBench (Zheng et al., 2024)), it does not scale automatically due to the heavy reliance on manual annotation. To reduce human

---

*Work done during the internship in Sea AI Lab. Email: k.duan@u.nus.edu

†Corresponding authors.

[1]The code is available at https://github.com/sail-sg/ActivePRM

efforts, Monte Carlo (M.C.) Estimation Methods (Wang et al., 2024; Wei et al., 2024; Luo et al., 2024) were proposed. However, these approaches come with high computational costs (massive rollouts are required for accurate estimation) and struggle to accurately identify the first error step (Zheng et al., 2024). To address this challenge, Qwen2.5-Math-PRM (Zhang et al., 2025) proposed using LLM-as-Judge — leveraging LLMs to detect the first error step — and filter out unreliable M.C. labels. It significantly boosts the performance of PRM on both ProcessBench (Zheng et al., 2024) and PRMBench (Song et al., 2025). More recently, UniversalPRM (Tan et al., 2025) relies solely on LLM-as-Judge with ensemble prompting (via majority voting), achieving new SOTA performance on ProcessBench within the same model size. However, the annotation costs are still considerable. We estimate the labeling costs of Qwen2.5-Math-PRM (Zhang et al., 2025) and UniversalPRM (Tan et al., 2025) and illustrate them in Figure 1. It shows that Qwen-Math-PRM-7B and UniversalPRM consume over $2^{34}$ and $2^{32}$ generated tokens, respectively. Refer to Appendix C for estimation strategy.

To reduce annotation costs, we propose ACTPRM, which uses a trained ensemble PRM to identify and select uncertain data for annotation by a high-capability reasoning model. Our approach trains a PRM with ensemble heads for uncertainty estimation. For each reasoning step, we compute the mean $\mu$ and standard deviation $\sigma$ of ensemble predictions, identifying uncertain steps when prediction confidence is outside threshold $1 - \delta_{pred} < \mu < \delta_{pred}$ or variation exceeds $\delta_{std}$. We consider a CoT trajectory uncertain **if any step up to and including the first predicted error meets these criteria**. By annotating only uncertain data and training exclusively on this subset, we significantly reduce labeling costs while maintaining PRM performance.

To validate the effectiveness and efficiency of ACTPRM, we conducted comprehensive experiments in multiple settings:

- **Pool-based Evaluation (Section 5.1):** Using 100K labeled samples, ACTPRM achieved performance comparable to full-data tuning while reducing annotation costs by 50%. It consistently outperformed random selection under identical budget constraints.

- **One-shot Active Learning (Section 5.2):** Starting with our pool-based model, we applied ACTPRM to select uncertain samples from 1M+ unlabeled CoT trajectories from NuminaMath (Li et al., 2024). After annotation and fine-tuning, we achieved new SOTA performance of 75.0% on ProcessBench. As shown in Figure 1, ACTPRM surpasses prior SOTA models with significantly lower costs—outperforming UniversalPRM (Tan et al., 2025) by 0.7% using only 20% of its annotation budget and exceeding Qwen2.5-Math-PRM-7B by 1.5% with just 6% of its annotation budget.

Our contributions are summarized as follows: ❶ We propose an uncertainty-aware active learning approach ACTPRM for PRM training that selectively annotates informative reasoning steps using ensemble-based uncertainty estimation, significantly reducing labeling costs while maintaining performance. ❷ ACTPRM achieves state-of-the-art results (75.0% on ProcessBench, 65.5% on PRMBench) while requiring only 20% of the annotation budget compared to prior SOTA method UniversalPRM. ❸ We release all trained models, datasets, and code to ensure reproducibility and facilitate community adoption.

## 2 Preliminaries

### 2.1 Process Reward Models

**Problem Formulation.** Given a math problem $q$ and a corresponding solution trajectory $s = [s_1, s_2, \ldots, s_n]$, where $s_i$ denotes $i$-th step., we require a PRM to identify the correctness of each step until a wrong step is identified. We only label the steps up to and including the first error step following prior works (Lightman et al., 2023; Zheng et al., 2024), since the afterward steps are genuinely difficult to define their correctness. As a result, in practice the labels for a solution trajectory are either $[1, 1, \ldots, 1]$ or $[1, 1, \ldots, 0]$. Then a PRM could be trained using the typical BCE loss:

$$\mathcal{L}_{BCE}(s, y|\theta) = -\frac{1}{|s|} \sum_{1}^{|s|} y_i \log(P_\theta(s_i|s_{[:i]}, q)) + (1 - y_i) \log(1 - P_\theta(s_i|s_{[:i]}, q)), \quad (1)$$

where $P_\theta$ is the PRM parameterized by $\theta$ and $s_{[:i]}$ denotes the steps before $s_i$. When using PRM for inference, we set a threshold $\delta$ (usually 0.5) to identify the first step that has a correctness probability $P_\theta(s_i|s_{[:i]}, q)$ less than $\delta$.

**PRM Implementation Details.** A typical PRM is built upon a pretrained generative LLM by replacing the causal language model head with a binary classification head that outputs the probability of correctness at corresponding token position. In practice, we solely need the prediction at the end of each reasoning step and thus a prediction mask is used to mask out the prediction at the other positions.

### 2.2 Uncertainty Estimation for Classification

**Aleatoric Uncertainty.** As aforementioned, a typical PRM $P_\theta$ is trained as a binary classification task. The simplest way to measure the uncertainty for it is to use aleatoric uncertainty (Valdenegro-Toro & Saromo, 2022):[2].

$$\text{Aleatoric Uncertainty} \propto P_\theta(s_i) \log P_\theta(s_i). \quad (2)$$

**Epistemic Uncertainty.** In addition, ensemble of models is also a common way to estimate epistemic uncertainty (Valdenegro-Toro & Saromo, 2022) by quantifying the disagreement among ensemble models. For example, Liang et al. (2022); Gleave & Irving (2022) use an ensemble of reward models to estimate uncertainty in preference learning. for process reward modeling, we could leverage the standard deviation of the ensemble predictions as the uncertainty estimation:

$$\text{Epistemic Uncertainty} \propto \text{Var}(\{P_\theta(s_i)\}), \quad (3)$$

where $\{P_\theta\}$ is a set of models. It is worth noting that employing an ensemble of heads built upon a shared backbone is a common strategy to mitigate computational costs. We empirically study the combination of aleatoric and epistemic uncertainty and find that they are complementary to each other. Experimental results are shown in Section 5.1.

## 3 Related Work

**Active Learning and Uncertainty Estimation.** Active learning has been widely explored in the alignment of LLMs. Several studies adopt an online bandit formulation, leveraging uncertainty-aware reward models (RMs) for active exploration in response selection (Mehta et al., 2023; Dwaracherla et al., 2024; Liu et al., 2024; Melo et al., 2024; Gleave & Irving, 2022). For instance, Mehta et al. (2023) and Dwaracherla et al. (2024) use ensemble-based LLM heads to estimate epistemic uncertainty, prioritizing data most informative for preference learning. Similarly, Melo et al. (2024) propose an acquisition function combining both entropy (aleatoric uncertainty) and epistemic uncertainty. Our work builds on these approaches, empirically evaluating the role of both uncertainty types in the context of process reward modeling. Beyond active learning, ensemble methods—such as those in Coste et al. (2024)—have also proven effective in mitigating reward hacking (Amodei et al., 2016).

**Process Reward Models.** Different from outcome rewards (e.g., verifiable rewards (DeepSeek-AI et al., 2025) for mathematical reasoning problems) that assign rewards based on the final outcome, process rewards are assigned based on the intermediate steps of the problem-solving process. For a question and a corresponding solution with several steps, a PRM provides finer-grained rewards for each step. Til current stage, process reward modeling contains two categories: (*i*) *Process Reward as Q-values* and (*ii*) *Process Reward as Judger*. The former one (Wang et al., 2024; Luo et al., 2024; Wei et al., 2024; Li & Li, 2024) regards the process reward as the Q-values of the steps that estimate the probability of the policy model

---

[2]For simplicity, we use $P_\theta(s_i)$ as the aleatoric probability for the $i$-th step in the solution trajectory, where the full representation is $p_\theta(s_i|s_{[:i]}, q)$ as in Equation 1

---

**Algorithm 1** PRM Active Learning with Cold Start.

---

1: // The difference with full data tuning is colored.
**Input:** Ensemble PRM $P_\theta$, dataset $\mathcal{D} = \{(q,s)\}$, uncertainty thresholds $\delta_{pred}$ and $\delta_{std}$, generative LLM $M$, batch size $B$, learning rate $\eta$
2: **for** $\mathcal{B} \subset \mathcal{D}$ **do**
3:     $P_\theta(\mathcal{B}) \leftarrow \text{Forward}(\mathcal{B})$
4:     $\widetilde{\mathcal{B}} = \{\}$
5:     **for** $(q,s) \in \mathcal{B}$ **do**
6:         **if** $\mathcal{U}_\theta^{\text{alea}}(s) \vee \mathcal{U}_\theta^{\text{epis}}(s)$ **then**                                   ▷ Equation 5
7:             $\widetilde{\mathcal{B}} \leftarrow \widetilde{\mathcal{B}} \cup \{(q,s)\}$
8:         **end if**
9:     **end for**
10:    $Y_{\widetilde{\mathcal{B}}} \leftarrow \text{labeling}(\widetilde{\mathcal{B}})$                        ▷ Labeling using generative LLM
11:    $\mathcal{L} \leftarrow \frac{1}{|\widetilde{\mathcal{B}}|} \sum_{(s,y)\in(\widetilde{\mathcal{B}},Y_{\widetilde{\mathcal{B}}})} \mathcal{L}(s,y)$                 ▷ Equation 4
12:    $\nabla_\theta \mathcal{L} \leftarrow \text{Backward}(\mathcal{L})$
13:    $\theta \leftarrow \theta - \eta \nabla_\theta \mathcal{L}$
14: **end for**
**Output:** $P_\theta$

---

to reach the final correct answer. Specifically, they leverage the policy model that generates the solution steps to do Monte Carlo Estimation for each step. The estimated Q-values are used as the process rewards. However, recent works (Zhang et al., 2025; Zheng et al., 2024) show that this kind of process reward modeling suffers from identifying the process errors because it highly depends on the policy model and has large bias with the ground truth distribution. In contrast, the latter one (Lightman et al., 2023; Zhang et al., 2025) regards the process reward model as a proxy for identifying the intermediate process errors and the corresponding trained model achieves better performance on several benchmarks (Zheng et al., 2024; Song et al., 2025). In this work, we follow the latter one and regard the process reward as a judge that tries to identify the first error steps in the solutions if any. In addition, there are other works related to PRM. For example, Yuan et al. (2024) tries to train a PRM with a fashion of outcome reward modeling (ORM). Cheng et al. (2025); Cui et al. (2025) proposed RL training frameworks that integrate PRM as finer-grained supervison.

## 4 Efficient Process Reward Labeling via Active Learning

Labeling the process rewards for a large-scale dataset is very expensive as it either requires human experts to annotate the correctness of each step for each solution as in the previous work (Lightman et al., 2023) or leverages highly capable generative models to imitate human experts (Zhang et al., 2025). Even though the latter one is automated, it is still resource-consuming since the test time scales up with the difficulty of math problems.

To mitigate this issue, we propose to leverage active learning to make the PRM proactively select the data that is most informative to train on. Specifically, we train a PRM with ensemble heads to enable uncertainty estimation following Liang et al. (2022); Gleave & Irving (2022). As shown in Algorithm 1, We forward the data candidates to the ensemble PRM (line 3) and estimate the prediction uncertainty for each data point (line 5-6). Then we omit the data that the ensemble PRM is confident about (line 7) and only label the other retained data with a generative reasoning LLM (line 10). Then, we only backpropagate from the loss of labeled data (line 11). By doing so, we could considerably reduce the labeling cost while maintaining the PRM performance. Now we introduce our two key differences with the original finetuning: *ensemble PRM training* and *uncertain data selection*.

**Ensemble PRM Training.** In this work, we use ensemble of PRMs to estimate the epistemic uncertainty following Gleave & Irving (2022); Liang et al. (2022). Specifically, we use a shared LLM backbone and build multiple binary classification heads on top of it. In our training, the diversity of ensemble models is ensured by two ways: (*i*) the random initialization of the head layers and (*ii*) a diversity regularization term (Dwaracherla et al., 2024): $\mathcal{L}_{\text{div}} = \lambda \cdot \frac{1}{n} \sum_{i=1}^n ||\phi^i - \phi^i_{\text{init}}||_2$, where $\{\phi^i\}$ are the parameters of the ensemble heads and n is the number of ensemble heads. It is a $L2$ term penalizing the distance between

Listing 1: Pseudo code of uncertainty estimation.

```
1  # Compute ensemble predictions (num_ensemble , num_step)
2  score = llm(input_ids)
3  means, stds = score.mean(0), score.std(0)  # Per-step statistics
4  # Equ. 5 left
5  epistemic_uncertainty = stds >= std_threshold
6  # Equ. 5 right
7  alearotic_uncertainty = (means <= pred_threshold) & (means >= 1 -
      pred_threshold)
8  # Alg. 1 line 6
9  uncertainty = any(epistemic_uncertainty | alearotic_uncertainty)
```

the ensemble head parameters and their initial parameters. Our training objective for the ensemble PRM is therefore formulated as follows

$$\mathcal{L}(y,s) = \frac{1}{n} \sum_{i=1}^{n} \left( \mathcal{L}_{BCE}(y,s|\theta, \phi^i) + \lambda ||\phi^i - \phi^i_{\text{init}}||_2 \right), \tag{4}$$

where $\theta$ denotes the backbone parameters and $\mathcal{L}_BCE$ is from Equation 1, that computes the loss for a certain head.

**Uncertain Data Selection.** Considering a batch of data candidates $\mathcal{D} = \{(q,s)\}$, we first forward the data to the ensemble PRM $P_\theta$ to get the ensemble predictions $P_\theta(\mathcal{D}) \in \mathbb{R}^{n \times |\mathcal{D}| \times |s|}$. for each data $(q,s) \in \mathcal{D}$, we could determine the hard-value aleatoric and epistemic uncertainty with pre-set thresholds. Briefly, the aleatoric (or epistemic) uncertainty is defined as 1 if uncertainty occurs at any step prior to the first predicted error step; otherwise, it is 0. A formal definition is as follows:

$$\mathcal{U}_\theta^{\text{alea}}(s) = \bigvee_{i=0}^{\mathcal{E}(s)} \left( \max \left( \mu(P_\theta(s_i)), 1 - \mu(P_\theta(s_i)) \right) < \delta_{pred} \right) ; \ \mathcal{U}_\theta^{\text{epis}}(s) = \bigvee_{i=0}^{\mathcal{E}(s)} \left( \sigma(P_\theta(s_i)) > \delta_{std} \right), \tag{5}$$

where $\mu(\cdot)$ and $\sigma(\cdot)$ are the mean and standard deviation of the ensemble predictions among ensemble heads and $\vee$ denotes the logical 'OR' operation. Moreover, the $\mathcal{E}(s)$ denotes the first error step in the solution trajectory $s$, defined as $\mathcal{E}(s) = \min\{j \mid \mu(s_j) < \delta\}$, where $\delta$ is the threshold for the correctness, typically set to 0.5. This is because we only care about the correctness of the steps before the first error step since it is genuinely difficult to define the correctness of the steps afterwards. For further illustration, we also provide the pseudo code of the uncertainty estimation as in Listing 1. By following the uncertainty estimation strategy, we retain the data in $\mathcal{D}$ that satisfies either $\mathcal{U}_\theta^{\text{alea}}$ or $\mathcal{U}_\theta^{\text{epis}}$ as $\widetilde{\mathcal{D}}$. Then we could leverage expensive generative LLMs as judger (Zheng et al., 2024) to label the retained data in $\widetilde{\mathcal{D}}$.

## 5 Experiments

In Section 5.1, we first validate ACTPRM in a pool-based active learning setting using 100K labeled samples, including ablation studies on our uncertainty estimation strategy. Based on the optimal configuration, we then scale up to 1M unlabeled samples in Section 5.2, further proving our pipeline's efficiency and effectiveness.

### 5.1 Pool-Based Active Learning

#### 5.1.1 Experimental Settings

To evaluate our active learning strategy's effectiveness, we first conduct experiments in an offline setting where ACTPRM iteratively selects the most informative examples from a large unlabeled pool as detailed in Algorithm 1. We establish a strong baseline by comparing

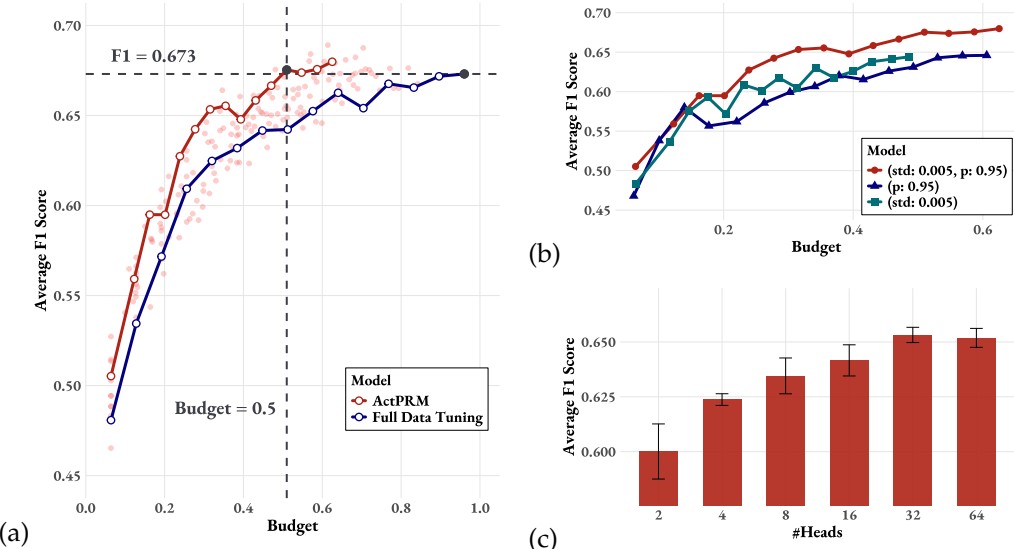

Figure 2: (a) Comparison of the average F1 score on ProcessBench between ACTPRM and random selection, plotted against the normalized budget positively correlated the number of labeled data instances consumed. The *semi-transparent* points represent all results in grid searching w.r.t. $\delta_{pred}$ and $\delta_{std}$. For the highlighted ACTPRM curve in the figure, $\delta_{pred} = 0.95$ and $\delta_{std} = 0.005$. (b) Ablation: uncertainty estimation strategies. (c) Ablation: number of ensemble PRM heads.

against full data tuning, where a model is trained on the complete dataset labeled by a single annotator. It is worth noting that as our data is randomly shuffled, the performance of full data tuning at intermediate training steps is essentially equivalent to the performance of random selection with the corresponding budget.

**Evaluation Benchmark.** We utilize ProcessBench (Zheng et al., 2024) to evaluate the effectiveness of PRMs. The test data in ProcessBench contains intermediate step errors and requires the PRM to identify the first error step. ProcessBench contains four subsets, and we report the average F1 score following the original work.

**Models.** We train ACTPRM based on Qwen2.5-Math-7B-Instruct.

**Training Dataset.** For dataset construction, we randomly select 100K data from Numinamath (Li et al., 2024) dataset after decontamination against the ProcessBench (Zheng et al., 2024) and PRMBench (Song et al., 2025). We leverage Qwen-2.5-Math-7B-Instruct to generate CoT reasoning trajectories for the selected data and further use QwQ-32B as a judge to annotate the process correctness for all trajectories following Zhang et al. (2025). For completeness, we provide the prompt template in Appendix A.

### 5.1.2 Experimental Results

**ACTPRM achieves comparable performance while reducing annotation costs by 50%.** We compare ACTPRM with full data tuning across a normalized budget, as illustrated in Figure 2 (a). The results demonstrate that ACTPRM achieves an average F1 score of 0.673 on ProcessBench, matching baseline performance while using only half the annotation budget. Furthermore, ACTPRM consistently outperforms random selection under the same budget constraints. Notably, at 50% budget, ACTPRM surpasses random selection by a significant margin of 3.3%. at the end of pool-based active training, ACTPRM achieves a better performance of 0.680 on ProcessBench while consuming solely 62.5% budget.

**ACTPRM Consistently Outperforms Random Selection Under Diverse $\delta_{pred}$ and $\delta_{std}$.** As shown in Figure 2 (a), the semi-transparent blue points represent all results of a grid searching over $\delta_{pred} \in \{0.9, 0.95, 0.97\}$ and $\delta_{std} \in \{0.01, 0.005, 0.002, 0.001\}$. One can see that most blue points are above the baseline (gray line) with the same budget, further demonstrating the effectiveness and robustness of ACTPRM.

**Ablation Study on Uncertainty Estimation Strategies.** We conduct an ablation study on uncertainty estimation strategies, i.e. using epistemic and aleatoric uncertainty. We selected the best setting ($\delta_{std} = 0.005, \delta_{pred} = 0.95$) searched by a grid search as in Figure 2 and ablates epistemic and aleatoric uncertainty by setting $\delta_{std} = $ inf and $\delta_{pred} = 0.5$, respectively. As shown illustrated in Figure 2 (b), solely use either epistemic or aleatoric uncertainty underperforms using both, indicating that epistemic and aleatoric uncertainty are complementary to each other.

**Ablation Study on Number of Heads for Ensemble PRM.** The number of heads for ensemble PRM controls how accurate our estimated epistemic uncertainty is. To find the trade-off between good estimation and computational overhead, we conduct an ablation study regarding it and show the results in Figure 2 (c), where we only consider epistemic uncertainty by setting $\delta_{std} = 0.005, \delta_{pred} = 0.5$ and report the averaged results with 3 runs. We empirically find that the performance continually grows with the number of heads and converges at about 32.

## 5.2 Achieving New SOTA Performance on ProcessBench (75.0%) with Solely 6% Annotation Cost.

Obtaining high-quality process supervision labels is costly. To demonstrate the efficiency of ACTPRM, we evaluate it in a one-shot active learning setting. Starting with the model trained in Section 5.1, we select the most uncertain samples from over 1M+ unlabeled examples and annotate them using a powerful reasoning model.

Figure 3 compares our estimated labeling costs with those of other real-world datasets for training PRMs, including MathShepherd (Wang et al., 2024), Consensus Filtering (Zhang et al., 2025), and Ensemble Prompting (Tan et al., 2025). Since the training data for Consensus Filtering is not publicly available, we estimate costs based on our data statistics. We introduce our estimation strategy in Appendix C.

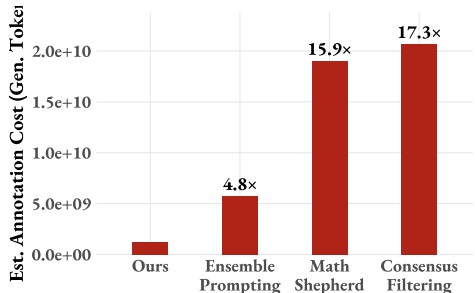

Figure 3: Estimated annotation costs (generated tokens) comparison between ACT-PRM and popular methods, including Ensemble Prompting (Tan et al., 2025), MathShepherd (Wang et al., 2024) and Consensus Filtering (Zhang et al., 2025).

Training a Qwen2.5-Math-7B-Instruct on our data, Ensemble Prompting data, MathShepherd data, and Consensus Filtering data yields ACTPRM, UniversalPRM, Qwen2.5-Math-7B-Math-Shepherd, and Qwen2.5-Math-PRM-7B in Table 1. We evaluated the performance of models trained on this labeled data on both ProcessBench and PRMBench benchmarks.

### 5.2.1 Experimental Settings

**Data Filtering with ACTPRM.** We used Qwen2.5-Math-7B-Instruct and Qwen2.5-Math-72B-Instruct to collect over 1 million (1,061,763) Chain-of-Thought (COT) trajectories from the Numinamath problem set (Li et al., 2024), after decontamination against the test benchmarks. ACTPRM was then applied to filter out high-confidence ($\delta_{pred} > 0.95$ and $\delta_{std} < 0.005$ following Section 5.1) data instances that were unnecessary for training, retaining the remaining data for labeling and training. This process resulted in a final dataset of 563,030 PRM data points labeled by QwQ-32B, reducing annotation costs by 47.0%.

**Models.** Obtaining the dataset, we continually train our ACTPRM in Section 5.1 on our filtered dataset. In addition, we empirically find that our retrained data is generally useful to other PRMs. Specifically, we also continually train Qwen2.5-Math-PRM-7B (the previous SOTA model on ProcessBench) on our constructed data. The resultant model is named ACTPRM-X[3].

---

[3]X stands for extended version.

| Models | GSM8K | MATH | Olympiad Bench | OmniMath | Average F1 |
|---|---|---|---|---|---|
| *LLM-as-judge* | | | | | |
| o1-Mini | 0.932 | 0.889 | 0.872 | 0.824 | 0.879 |
| Deepseek-R1-Distill-32B | 0.817 | 0.739 | 0.659 | 0.585 | 0.700 |
| QwQ-32B | 0.871 | 0.834 | 0.787 | 0.771 | 0.816 |
| *Process Reward Models (72B)* | | | | | |
| Qwen2.5-Math-PRM-72B | 0.873 | 0.806 | 0.743 | 0.711 | 0.783 |
| *Process Reward Models (7B+)* | | | | | |
| Math-Shepherd-PRM-7B | 0.479 | 0.295 | 0.248 | 0.238 | 0.315 |
| Qwen2.5-Math-7B-Math-Shepherd | 0.625 | 0.316 | 0.137 | 0.077 | 0.289 |
| EurusPRM-Stage2 | 0.473 | 0.357 | 0.212 | 0.209 | 0.313 |
| Qwen2.5-Math-7B-PRM800K | 0.683 | 0.626 | 0.507 | 0.443 | 0.565 |
| Ensemble-PRM-PRM800K (ours) | 0.705 | 0.630 | 0.472 | 0.433 | 0.560 |
| PURE-PRM-7B | 0.690 | 0.665 | 0.484 | 0.459 | 0.575 |
| Qwen2.5-Math-PRM-7B | 0.824 | 0.776 | 0.675 | 0.663 | 0.735 |
| Universal-PRM | **0.858** | 0.777 | 0.676 | 0.664 | 0.743 |
| ACTPRM (ours) | 0.816 | 0.798 | 0.714 | 0.670 | 0.750 |
| ACTPRM-X (ours) | 0.827 | **0.820** | **0.720** | **0.673** | **0.760** |

Table 1: Performance comparison on ProcessBench. We report the results in the same calculation method with ProcessBench. $^\diamond$ denotes the results are from Qwen PRM's report (Zhang et al., 2025).

**Benchmarks.** We use ProcessBench (Zheng et al., 2024) and PRMBench (Song et al., 2025) to evaluate the effectiveness of our trained model. Different from ProcessBench that collects intermediate errors from real-world generative models, PRMBench heuristically builds intermediate errors by manipulating correct steps.

**Baselines.** We compare with the following PRMs: ❶ *Qwen2.5-Math-PRM-7B* (Zhang et al., 2025): This model uses consensus filtering for labeling. It labels 860K data twice using two methods (LLM-as-judge [Zheng et al., 2024] and Mathshepherd [Wang et al., 2024]) and filters out 40% of the data where the labels disagree. ❷ *Pure-PRM-7B* (Cheng et al., 2025): A Qwen2.5-Math-based PRM trained on PRM800K using a two-stage strategy: warming up the PRM head and then fine-tuning the entire model. ❸: *EurusPRM-Stage2* (Cui et al., 2025): A PRM resulting from the Implicit PRM approach (Yuan et al., 2024), which derives process rewards from an ORM. ❹ *Universal-PRM* (Tan et al., 2025): A Qwen2.5-Math-based model trained with data augmentation techniques like ensemble prompting and reverse verification. ❺ *Math-Shepherd-PRM-7B* (Wang et al., 2024): a PRM trained on process labels that estimates hard Q-values for the policy model. ❻ *Qwen2.5-Math-7B-Math-Shepherd* (Zhang et al., 2025): a PRM trained on 860K data labeled using MathShepherd. ❼ *Ensemble-PRM-PRM800K (ours)*: a model with ensemble heads trained by ourselves on PRM800K without active learning.

### 5.2.2 Experimental Results

**ACTPRM and ACTPRM-X achieve new SOTA performance on ProcessBench compared with same size models.** The evaluation results on ProcessBench are shown in Table 1. ACTPRM achieves an average F1 score of 0.750, outperforming Qwen2.5-Math-PRM-7B by a margin of 1.5%. Furthermore, ACTPRM-X training based on Qwen2.5-Math-PRM-7B achieves a new SOTA performance on ProcessBench with an average F1 of 0.760, outperforming the second-place model (Universal-PRM) with a margin of 1.7% and improve the performance of Qwen2.5-Math-PRM-7B by a significant margin of 2.5%.

**QwQ-32B (our PRM label annotator) outperforms all PRMs on ProcessBench.** As shown in Table 1, QwQ-32B outperforms all PRMs including 72B models. It indicates the reliability of utilizing QwQ-32B as a PRM label annotator as it provides a high empirical upperbound for the training PRMs.

**ACTPRM-X achieves new SOTA performance on PRMBench, on-par with GPT-4o.** We further test our models on PRMBench and show the results in Table 2. As on the leaderboard, ACTPRM achieves the best performance within 7B PRMs and ACTPRM-X achieves new

| # | Models | Simlicity | Soundness | Sensitivity | Average |
|---|--------|-----------|-----------|-------------|---------|
| | *LLM-as-judge* | | | | |
| 1 | Gemini-2.0-thinking-exp-1219 | 0.662 | 0.718 | 0.753 | 0.688 |
| 1 | o1-mini | 0.646 | 0.721 | 0.755 | 0.688 |
| 4 | GPT-4o | 0.597 | 0.709 | 0.758 | 0.668 |
| 6 | Gemini-2.0-flash-exp | 0.627 | 0.673 | 0.754 | 0.660 |
| | *Process Reward Models (72B)* | | | | |
| 3 | Qwen-2.5-Math-PRM-72B | 0.546 | 0.739 | 0.770 | 0.682 |
| | *Process Reward Models (7B+)* | | | | |
| 7 | Qwen2.5-Math-PRM-7B | 0.521 | 0.710 | 0.755 | 0.655 |
| 9 | Pure-PRM-7B | 0.522 | 0.702 | **0.758** | 0.653 |
| 7 | ACTPRM (ours) | 0.536 | 0.713 | 0.752 | 0.655 |
| 5 | ACTPRM-X (ours) | **0.545** | **0.727** | 0.756 | **0.667** |

Table 2: Performance comparison on PRMBench. All results of the other models are from the official leaderboard, "-" denotes the ranking.

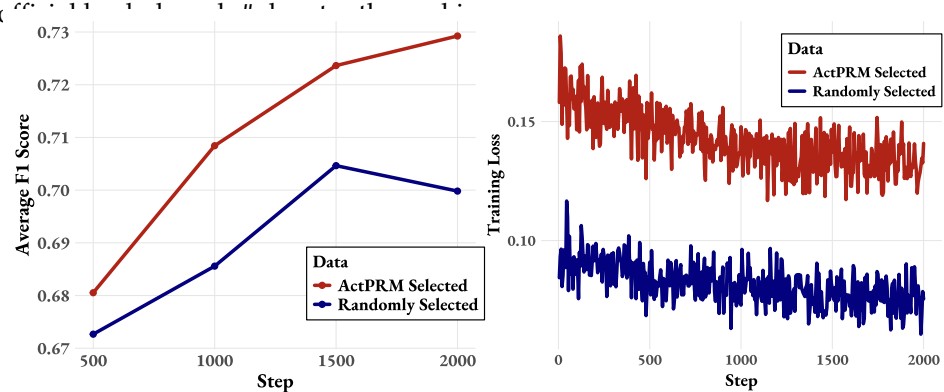

Figure 4: ProcessBench performance (*left*) and training loss (*right*): ActPRM v.s. random data selection on 1M NuminaMath Rollouts.

SOTA performance (0.667), outperforming the other models by a large margin of at least 1.2% and on-par with GPT-4o (OpenAI et al., 2024a).

### 5.2.3 Comparative Experiment with Random Selection

A potential concern is that while ACTPRM achieves state-of-the-art (SOTA) performance on several benchmarks, this success might be attributed solely to the high quality of our collected data pool, rather than the method itself. To address this, we conducted a comparative study with random selection. Specifically, we randomly selected 256K data points from our retained dataset as the experimental group. For the control group, we randomly selected the same number of data points from the entire data pool (over 1M) and used the same annotator to label any unlabeled data (i.e., data not in the retained set). We then continually train ACTPRM checkpoint, as in Sec 5.1, on both datasets. The results, including performance on ProcessBench and training loss, are shown in Figure 4.

**ACTPRM outperforms random selection of the same amount of data.** As illustrated in Figure 4 (left), the model trained on data selected by ACTPRM consistently achieves significantly better results than the model trained on randomly selected data. To further validate this, we compare their training losses in Figure 4 (right). The model trained on ACTPRM-selected data exhibits a consistently higher training loss, with a margin of 0.05, suggesting that the data selected by ACTPRM is more challenging and informative, thereby enhancing the learning process.

## 6 Conclusion and Future Work

In this work, we address the high annotation costs associated with training Process Reward Models (PRMs) by proposing ACTPRM, an uncertainty-aware active learning framework

that selectively annotates the most informative reasoning steps. By leveraging an ensemble PRM to estimate uncertainty and strategically labeling only uncertain data, ACTPRM significantly reduces annotation costs while maintaining competitive performance. Extensive experiments demonstrate that ACTPRM achieves a new state-of-the-art (75.0% on Process-Bench) with merely at most 20% of the labeling budget required by prior methods. Our results highlight the potential of efficient data selection for scalable PRM training, and we commit to releasing all models, datasets, and code to foster further research in this direction.

To further enhance PRM's performance, several promising directions can be explored. First, leveraging larger base models and more advanced LLM judges (e.g., O1-mini) could yield significant improvements. Second, implementing the framework in an online setting would ultimately enable PRM to iteratively refine its performance through active learning. Additionally, integrating online PRM training with reinforcement learning frameworks—such as actor-critic methods—presents an exciting avenue for research.

## Acknowledgement

This project was partly supported by the MOE AcRF Tier 1 grant with grant number 251RES2514.

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

## A LLM-as-Judger Prompt Template

For LLM-as-Judger, we follow the prompt in Zhang et al. (2025).

```
1  I will provide a math problem along with a solution. They will be formatted as follows:
2  [Math Problem]
3  <math_problem>
4  ...(math problem)...
5  </math_problem>
6  [Solution]
7  <paragraph_1>
8  ...(paragraph 1 of solution)...
9  </paragraph_1>
10 ...
11 <paragraph_n>
12 ...(paragraph n of solution)...
13 </paragraph_n>
14
15 Your task is to review each paragraph of the solution in sequence, analyzing,
16 verifying, and critiquing the reasoning in detail. You need to provide the
17 analyses and the conclusion in the following format:
18 <analysis_1>
19 ...(analysis of paragraph 1)...
20 </analysis_1>
21 ...
22 <analysis_n>
23 ...(analysis of paragraph n)...
24 </analysis_n>
25 <conclusion>
26 Correct/Incorrect
27 </conclusion>
28
29 * When you analyze each paragraph, you should use proper verification, recalculation, or
        ↪ reflection to indicate whether it is logically and mathematically valid. Please
        ↪ elaborate on the analysis process carefully.
30 * If an error is detected in any paragraph, you should describe the nature and cause of the
        ↪ error in detail, and suggest how to correct the error or the correct approach. Once a
        ↪ paragraph is found to contain any error, stop further analysis of subsequent
        ↪ paragraphs (as they may depend on the identified error) and directly provide the
        ↪ conclusion of "Incorrect." For instance, given a solution of five paragraphs, if an
        ↪ error is found in the third paragraph, you should reply in the following format:
31 <analysis_1>
32 ...(analysis of paragraph 1)...
33 </analysis_1>
34 <analysis_2>
35 ...(analysis of paragraph 2)...
36 </analysis_3>
37 <analysis_3>
38 ...(analysis of paragraph 3; since an error is found here, also provide detailed critique and
        ↪ correction guideline)...
39 </analysis_3>
40 <conclusion>
41 Incorrect
42 </conclusion>
43 Note that the analyses of paragraphs 4 and 5 should be skipped as the paragraph 3 has been
        ↪ found to contain an error.
44 * Respond with your analyses and conclusion directly.
45 -----------------------------------------------
46 The following is the math problem and the solution for your task:
47 [Math Problem]
48 {tagged_problem}
49 [Solution]
50 {tagged_response}
```

## B More Experiment Results

### B.1 Problem diversity is important for Training PRMs

PRM800K (Lightman et al., 2023) is a widely used and human-annotated dataset for PRM training, which contains 800K step-level labels across 75K tree-of-thoughts solutions to 12K MATH (Hendrycks et al., 2021). Our empirical results show that models trained on our dataset (100K samples from 100K diverse questions) consistently and significantly outper-

| | # Problem set | # CoT Trajectories | ProcessBench F1 score |
|---|---|---|---|
| **PRM800K** | 7,500 | 460,000 | 0.575 |
| **NuminaMath (Random Selected)** | 100,000 | 100,000 | 0.673 |

Table 3: Comparison between PRM800K and 100K data collected from NuminaMath labeled by Qwen-QwQ.

form those trained on PRM800K[4] (369K samples from only 12K questions) on ProcessBench. These findings suggest that problem diversity plays a more crucial role in PRM training than the number of step-level annotations.

## C Annotation Cost Estimation

We estimate the labeling cost based on the statistics of our 1M data collected from Numina-Math Li et al. (2024) using Qwen2.5-Math-7B-Instruct and Qwen2.5-Math-72B-Instruct. We introduce the statistics in Table 4.

| | Value | Source |
|---|---|---|
| **# Reasoning Steps ($S$)** | 8.845 | Qwen Models' rollouts |
| **# Tokens per Rollout ($R$)** | 625.098 | Qwen Models' rollouts |
| **# Tokens per Critic Response from Judge ($C$)** | 1,919.860 | Qwen-QwQ's responses as LLM-as-Judge |

Table 4: Statistics of 1M NuminaMath CoT Trajectories collected by Qwen2.5-Math Models.

In addition to the statistics, we also use $N$ to denote the data number of the dataset and show this statistic of each model's training dataset in Table 5

| Dataset | # Labeled Data |
|---|---|
| **ACTPRM** | 624,000 (labeled in two stages) |
| **Qwen2.5-Math-PRM-Math-shepherd** | 860,000 |
| **Qwen2.5-Math-PRM** | 860,000 |
| **UniversalPRM** | 690,000 |

Table 5: Data number of datasets.

Using the statistics, we compute the estimated labeling cost for ACTPRM, Qwen2.5-Math-PRM-Math-shepherd (Zhang et al., 2025), Qwen2.5-Math-PRM(Zhang et al., 2025), UniversalPRM Tan et al. (2025) as follows:

- Qwen2.5-Math-PRM-Math-shepherd: $N \times S \times 8 \times R/2$, where 8 is the number of rollouts per step set in Zhang et al. (2025). We divided by two since the number of tokens for rollouts varies based on the position of reasoning step. For latter reasoning step, it requires less reasoning tokens. As a result, the expectation of tokens per rollout should be half of the number of tokens of the complete rollout.

- Qwen2.5-Math-PRM: $N \times S \times 8 \times R/2 + N * C$. It used consensus filtering for each data, the cost is both from MathShepherd ($S \times 8 \times R/2$) and LLM-as-Judge ($C$).

- UniversalPRM: $N \times C \times 4 + N \times S$, where 4 is the number of ensemble prompts from the original paper (Tan et al., 2025) and another $N \times S$ is for its semantic-based step seperation.

- ACTPRM: $N \times C$. We solely use Qwen-QwQ as Judge and do not include any other operations.

---

[4]https://huggingface.co/datasets/HuggingFaceH4/prm800k-trl-dedup

