# OpenReview forum: "Efficient Process Reward Model Training via Active Learning"
_colmweb.org/COLM/2025/Conference — COLM 2025_

### Official Review · Reviewer_hjfr · 2025-04-26

**Rating:** 7
**Confidence:** 4
**Ethics Flag:** 1

**Summary:**

This paper proposes ACTPRM, an active learning method for training Process Reward Models (PRMs) efficiently. Instead of labeling all steps in reasoning trajectories, ACTPRM selectively queries "uncertain" examples (identified using ensemble-based epistemic and aleatoric uncertainty) for labeling, dramatically reducing annotation costs. Experiments show ACTPRM matches or exceeds previous state-of-the-art performance (e.g., 75% on ProcessBench) with only 6–20% of the labeling cost.

**Reasons To Accept:**

Reducing annotation costs is critical for scaling PRMs, especially in math reasoning tasks.

The paper explains ensemble head setup, uncertainty metrics, and selection strategies very cleanly (Algorithm 1 is crisp and actionable).

Across ProcessBench and PRMBench, ACTPRM and ACTPRM-X achieve new SOTA with dramatically lower labeling costs.

Includes ablations on uncertainty types, ensemble size, budget-efficiency curves, and comparative studies against random selection.

The paper is easy to follow, well-structured, and offers thoughtful discussion of limitations and future work.

**Reasons To Reject:**

The idea of using ensemble uncertainty for active learning is well-established (e.g., in preference learning, alignment studies). The technical novelty mainly lies in applying this idea to PRM training rather than inventing new uncertainty quantification techniques.

The method assumes availability of very strong LLMs (e.g., QwQ-32B) for labeling the uncertain examples. In weaker resource settings, ACTPRM's performance may degrade.

The uncertainty thresholds are tuned once and reused. It’s unclear how sensitive the method is to these choices across very different domains (e.g., non-math CoT).

The paper could better discuss ethical implications (e.g., reliance on large models, environmental costs of retraining, biases in LLM judges).

---

> ### Author Response · Authors · 2025-06-02
> **Response**
>
> Thank you for your thoughtful and encouraging review. Below we respond to the comments in **Weaknesses (W)** and **Questions (Q)**.
>
> ---
>
> **W1**: The technical novelty mainly lies in applying this idea to PRM training rather than inventing new uncertainty quantification techniques.
>
> We agree active learning is established. However, applying it to sequential CoT data presents unique challenges:
>
> - Step-wise uncertainty estimation for process reward modeling remains under-explored
>
> - Our method achieves SOTA PRM performance with only 20% data, demonstrating its novel efficiency for process reward modeling.
>
> ---
>
> **W2**: The method assumes availability of very strong LLMs (e.g., QwQ-32B) for labeling the uncertain examples. In weaker resource settings, ACTPRM's performance may degrade.
>
> Thanks for the comments and we hope to further clarify our research goal here. Our research objective is about how to efficiently sample data for querying any labeling oracle to train process reward models. While we use QwQ-32B here, oracle choice is orthogonal from our core goal of uncertainty-based sampling.
>
> ---
>
> **W3**: The uncertainty thresholds are tuned once and reused. It’s unclear how sensitive the method is to these choices across very different domains (e.g., non-math CoT)
>
> In Figure 2a, we provide all results of grid search w.r.t. the hyper-paramters ($\delta_{std} \in \{0.9, 0.95, 0.97\}$ and $\delta_{pred} \in \{​​0.01, 0.005, 0.002, 0.001\}$) with semi-transparent points. The results show that most of the results are better than the baseline (full data tuning with same data budget), indicating that the method is not sensitive to the choice of hyperparameters.
>
> For cross domain testing, we conjecture that the hyper-paramter would remain stable and still need further demonstration. In response to the question, we are willing to extend our active learning framework to code generation for validation, we propose:
>
> - Dataset Adaptation: Curate code-specific datasets with step-wise annotations (e.g., HumanEval+ problem set).
>
> - PRM Training: Train a code-oriented PRM using our acquisition strategy.
>
> - Benchmarking: Test performance (e.g., pass@k) on code tasks and compare against [1].
>
> ---
>
>
> **W4**: The paper could better discuss ethical implications (e.g., reliance on large models, environmental costs of retraining, biases in LLM judges).
>
> Thank you for the valuable feedback. While our work focuses on annotation efficiency, we acknowledge the ethical concerns raised. ACTPRM reduces reliance on large models by minimizing their usage through uncertainty-based selection, lowering environmental costs by cutting annotation overhead by up to 80%, and using shared backbones for efficiency. We will add these points to the revision to address broader implications.
>
> ---
>
> [1] Dai, Ning, et al. "Process supervision-guided policy optimization for code generation." arXiv preprint arXiv:2410.17621 (2024).

---

> > ### Author Response · Authors · 2025-06-09
> >
> > Dear Reviewer hjfr,
> >
> > Thank you for your time and review—we’d truly appreciate it if you could take a moment to review our rebuttal if you have a chance. Your insights are invaluable to us.
> >
> > Best regards,
> > Authors

---

> > ### Comment · Reviewer_hjfr · 2025-06-10
> >
> > Thank you for the details!

---

### Official Review · Reviewer_YDbb · 2025-05-11

**Rating:** 6
**Confidence:** 3
**Ethics Flag:** 1

**Summary:**

The paper proposes ACTPRM, an active learning framework for training Process Reward Models (PRMs) that significantly reduces annotation costs while maintaining or improving performance. By leveraging ensemble-based uncertainty estimation, ActPRM selectively annotates the most uncertain reasoning steps, achieving a 50% reduction in labeling costs compared to full-data tuning.

**Questions To Authors:**

1. In Section 5.1, it remains unclear whether ACTPRM ispre-trained before executing Algorithm 1, which I think is not clearly outlined in paper. If ACTPRM is pre-trained, I hope to see more details. If it is initialized directly from the base model Qwen2.5-Math-7B-Instruct, this raises another question: How much data would be labeled as "uncertain"? Intuitively, this proportion should be substantial since the ACTPRM lacks prior knowledge, implying most of the data (100K from Numinamath) would require annotation.
2. Lines 197 and 205 contain ambiguous phrasing. Active learning should operate on a large unlabeled pool—how should Line 197 be interpreted in this context?
3. Do authors conduct further analysis of the uncertain data metrics (aleatoric and epistemic)? Why are these specific metrics chosen? Are other factors considered?
4. I have questions about the initialization method for the ensemble heads. Do authors explore alternative approaches, such as averaging the final layer embeddings across the vocab dimension (a conventional practice)? Given that 32 heads are ultimately selected, could the authors provide insights into why so much heads is necessary?
5. In Section 5.2, the authors use QwQ-32B to annotate the PRM dataset, yet in the final results, QwQ-32B outperforms all PRMs on ProcessBench. This suggests potential unresolved influence from data distillation via QwQ. Is this an inherent limitation of active learning? A deeper analysis could be valuable.
6. How should Line 190 be interpreted? Can uncertainty scores determine the correctness of a given step?
7. Typo Error: Line 179, L_BCE.

**Reasons To Accept:**

1. This paper proposes a novel active learning approach (ACTPRM) for PRM training that effectively addresses the critical challenge of high annotation costs.
2. The experimental results demonstrate significant advantages, achieving state-of-the-art (SOTA) performance on both ProcessBench and PRMBench.
3. The ablation studies are comprehensive, employing a two-stage experimental design to identify optimal hyperparameters.

**Reasons To Reject:**

1. The authors could provide a more coherent explanation and clarification regarding the derivation and definition of the uncertainty score in Section 4.
2. The writing could be further refined for clarity, particularly in detailing the active learning experimental procedure and highlighting the progressive relationship between Sections 5.1 and 5.2.

---

> ### Author Response · Authors · 2025-06-02
> **Response**
>
> Thank you for your thoughtful review. We sincerely appreciate your specific suggestions regarding typographical corrections, which we will incorporate into the revised manuscript. Below we respond to the comments in **Weaknesses (W)** and **Questions (Q)**.
>
> ---
>
> **W1**: The authors could provide a more coherent explanation and clarification regarding the derivation and definition of the uncertainty score in Section 4
>
> We will enhance Section 4's clarity by adding pseudocode to illustrate Equation 5:
>
> ```python
> # Compute ensemble predictions (num_ensemble, num_step)
> score = llm(input_ids)
> means, stds = score.mean(0), score.std(0)  # Per-step statistics
> # Equ. 5 left
> epistemic_uncertainty = stds >= std_threshold
> # Equ. 5 right
> alearotic_uncertainty = (means <= pred_threshold) & (means >= 1 - pred_threshold)
> # Alg. 1 line 6
> uncertainty = any(epistemic_uncertainty | alearotic_uncertainty)
> ```
>
>   ---
>
> **W2**: The writing could be further refined for clarity, particularly in detailing the active learning experimental procedure and highlighting the progressive relationship between Sections 5.1 and 5.2.
>
> We will clarify the progression: Section 5.1 validates ACTPRM in pool-based active learning (100K labeled samples), including uncertainty strategy ablations. Section 5.2 then scales this optimal configuration to 1M unlabeled samples, demonstrating pipeline efficiency (lines 196-199).
>
> ---
>
> **Q1**: Is ActPRM pre-trained or initialized directly from the base model? If it is directly initialized, how much data would be labeled as ‘uncertain’?
>
> The ActPRM is directly initialized from the base model while the ensemble heads are randomly initialized. ActPRM allows a cold start: for the initial stage, all data will automatically be considered as uncertain and after a number of steps, the trust percentage will gradually increase and stabilize in the end. An illustration of this procedure is anonymously presented at [trust-percentage.png](https://postimg.cc/VrqhJ3Rs). As illustrated in Figure 2a, ActPRM finally consumed 60% data and saved 40% data by labeling them as confident points.
>
> ---
>
> **Q2**: Lines 197 and 205 contain ambiguous phrasing. Active learning should operate on a large unlabeled pool—how should Line 197 be interpreted in this context?
>
> The line 197 refers to the setting of ‘pool-based active learning’ where all data are labeled ahead. However, the active learning method will only use the data labels which are uncertain.
>
> ---
>
> **Q3**: Do authors conduct further analysis of the uncertain data metrics (aleatoric and epistemic)? Why are these specific metrics chosen? Are other factors considered?
>
> Our choice of aleatoric (data) uncertainty and epistemic (model) uncertainty follows established practices in active learning literature [1,2]. These metrics capture complementary aspects of model uncertainty: Aleatoric addresses inherent data noise, while epistemic identifies knowledge gaps in the model. As a result, we did not consider other factors.
>
> ---
>
> **Q4**: Do authors explore alternative initialization approaches, such as averaging the final layer embeddings across the vocab dimension (a conventional practice)?
>
> Thank you for the suggestion. Ensemble heads require distinct random initializations to measure epistemic uncertainty via standard deviation. Averaging final-layer embeddings would yield identical deterministic initialization across heads, conflicting with this objective.
>
> ---
>
> **Q5**: Given that 32 heads are ultimately selected, could the authors provide insights into why so many heads are necessary?
>
> Section 5.1.2 (Figure 2c) shows performance growing with head count and converging at 32. It is worth noting that the ensemble heads (three layer MLPs) are lightweight for training and impose minimal training overhead.
>
> ---
>
> **Q6**: Is it an inherent limitation of active learning that QwQ-32B outperforms all PRMs on ProcessBench?
>
> Thank you for the question. We view this not as an inherent limitation of active learning. The goal of active learning is about how to efficiently query an oracle (e.g., QwQ-32B) to train a PRM but not to bridge the gap between the training model and the oracle model (data annotator).
>
> We also note that the observed performance gap between our trained model and QwQ-32B  may also be attributed to the following factors:
>
> - QwQ-32B is used as a generative reward model while PRM is an discriminative model.
>
> - The trained PRM is smaller in size.
>
> ---
>
> **Q7**: How should Line 190 be interpreted? Can uncertainty scores determine the correctness of a given step?
>
> The mean ($\mu(s_j)$) serves a dual purpose: (1) as the final prediction, and (2) for aleatoric uncertainty estimation via $\max(1-\mu, \mu) < \delta_{pred}$. We will clarify this ambiguity in revision.
>
> ---
>
> [1] Sample-Efficient Reinforcement Learning from Human Feedback via Information-Directed Sampling. Arxiv.
>
> [2] Efficient exploration for llms. PMLR.

---

> > ### Author Response · Authors · 2025-06-09
> >
> > Dear Reviewer YDbb,
> >
> > Thank you for your time and review—we’d truly appreciate it if you could take a moment to review our rebuttal if you have a chance. Your insights are invaluable to us.
> >
> > Best regards,
> > Authors

---

> > > ### Comment · Reviewer_YDbb · 2025-06-10
> > >
> > > Thank you for your detailed reply, it addressed my question.

---

### Official Review · Reviewer_ZZ8c · 2025-05-14

**Rating:** 8
**Confidence:** 4
**Ethics Flag:** 1

**Summary:**

The paper introduces a novel active learning framework for training Process Reward Models using uncertainty-aware data selection. The proposed method uses ensemble heads to estimate both aleatoric and epistemic uncertainty (rare distinction in ML literature) and only annotate uncertain samples using a strong model as a judge. Reported experiments show that the proposed method achieves SOTA performance on ProcessBench and PRMBench, outperforming existing baselines while using only a small percentage of the annotation budget. The paper includes detailed robustness evaluations, and shows commitment to reproducibility by claiming to release code and models.

**Questions To Authors:**

Have you tried this on problems other than math reasoning? E.g. code generation. I expect it would transfer well, but it would be great to see empirical confirmation.

**Reasons To Accept:**

- The paper tackles an impactful problem - annotation costs in PRM training are very high, which presents a major challenge in tuning LLMs for multi-step reasoning.
- The use of both epistemic and aleatoric uncertainty through ensemble-based modeling is effective and is theoretically grounded.
- Strong empirical results
- Robust analysis

**Reasons To Reject:**

- I see no reasons to reject this paper.

---

> ### Author Response · Authors · 2025-06-02
> **Response**
>
> Thank you for your thoughtful and encouraging review. Below we respond to the comments in **Question (Q)**.
>
>   ---
>
> **Q1**: Have the authors tried this on problems, e.g., code generation, other than math reasoning? It would be great to see empirical confirmation.
>
> Thank you for the valuable suggestion. While our paper focuses on math reasoning—where process-based rewards are well-defined and data is readily available—we acknowledge the importance of broader applications like code generation.
>
> Although our current method is not evaluated on code generation (due to domain-specific training corpora and step granularity challenges), we conducted a preliminary literature survey to bridge this gap. For instance, [1] successfully applied process reward models (PRMs) to code generation via RL, validating the potential of PRMs in this domain.
>
> Future Work Plan:
> To extend our active learning framework to code generation, we propose:
>
> 1. Dataset Adaptation: Curate code-specific datasets with step-wise annotations (e.g., HumanEval+ problem set).
>
> 2. PRM Training: Train a code-oriented PRM using our acquisition strategy.
>
> 3. Benchmarking: Test performance (e.g., pass@k) on code tasks and compare against [1].
>
>
> We agree this is a promising direction and will prioritize it in follow-up studies.
>
> [1] Dai, Ning, et al. "Process supervision-guided policy optimization for code generation." arXiv preprint arXiv:2410.17621 (2024).

---

> > ### Author Response · Authors · 2025-06-09
> >
> > Dear Reviewer ZZ8c,
> >
> > Thank you for your time and review—we’d truly appreciate it if you could take a moment to review our rebuttal if you have a chance. Your insights are invaluable to us.
> >
> > Best regards,
> > Authors

---

> > ### Comment · Reviewer_ZZ8c · 2025-06-09
> >
> > Authors' comments address my questions.

---

### Official Review · Reviewer_p2Cc · 2025-05-16

**Rating:** 6
**Confidence:** 2
**Ethics Flag:** 1

**Summary:**

This paper proposes to use active learning for PRMs (Process Reward Models). An ensemble PRM is used. If the mean prediction confidence of the ensemble does not exceed a confidence threshold (0.95 worked best in the paper's results) for any step in a solution trajectory or if the standard deviation exceeds a threshold (0.005 worked best in the paper's results) for any step in a solution trajectory, then the math problem and the solution trajectory is selected to be labeled by a capable but costly LLM (Large Language Model) and used to train the PRM. The diversity of the ensemble models is fostered by using random initialization of the head layers and a diversity regularization term in the loss function for training the ensemble PRM. The proposed new approach is compared with existing approaches on ProcessBench and PRMBench (see Tables 1 and 2).  Experimental results show that the proposed method has relatively strong performance with significantly less data needing to be labeled by the capable but costly LLM. Additional experimental results are reported showing results with different confidence and standard deviation threshold values, {0.9, 0.95, 0.97} and {0.01, 0.005, 0.002, 0.001}, respectively. Also, an ablation study is done checking if the two types of uncertainty are complementary and they are found to be complementary. Also, experimental results are reported with different numbers of heads for the ensemble and it is found that performance improves as the number of heads grows until converging at about 32 heads.

Some parts of the paper were not clearly written. See below sections of review for details and examples.

**Questions To Authors:**

See above sections of review.

Also, I think the ablation study on number of heads for ensemble PRM would make sense to run with both epistemic and aleatoric uncertainty used for sampling since that is the actual best system that is being proposed in which the ensemble would be used. The existing experiment with only epistemic uncertainty considered is perhaps of some theoretical interest, but it could be that in practice the convergence properties are different when both aleatoric and epistemic uncertainty are considered, which is the suggested method, so testing with both would seem to be more practically useful and relevant.

I suggest expanding all acronyms before first use.

I suggest fixing the subscript on the reference to the loss function in line 179.

In Figure 2a, why does the curve for the ActPRM model stop at Budget = approximately 0.6?  Could it be run out all the way to Budget = 1.0?

In line 182 the variable n is used. It has been used previously in line 93 as the number of solution steps in a solution trajectory and in line 176 as the number of ensemble heads. I suppose in line 182 it means the number of ensemble heads and most readers will be able to figure this out, but it would be nice to make it clear to readers by perhaps using different variable names for these two different meanings.

In Algorithm 1, in the "Input" section a "generative LLM M" is mentioned, but it seems M is never used anywhere in the rest of the algorithm. In line 10, it might make sense to indicate the labeling is done by M.

In table 2, I guess the column "Simlicity" should be "Simplicity"?

It seems that the \delta_{pred} and \delta_{std} parameters were set to their best values via running through grid search over runs with a bunch of possible values. Does this give the proposed method an unfair advantage over other competing methods in the experimental results that are reported?  Is it suggested that the values of 0.95 (confidence threshold) and 0.005 (standard deviation threshold) should now always be used with the proposed new method, even for new tasks and applications of the method beyond the datasets used in the paper's experiments?  If not, how should those parameters be set for uses of the proposed method for new tasks and applications beyond the datasets used in the paper's experiments?

**Reasons To Accept:**

Seems to be novel if it's indeed the first work to use active learning to improve the data efficiency of training PRM.

New proposed method seems to be effective in experimental results reported.

**Reasons To Reject:**

Some parts of the paper were not clearly written, especially with regard to understanding the contents of the experimental results and how to interpret the results.

Some parts of the evaluation were difficult to understand. For example, for the results in Table 1, it was difficult to determine what the annotation cost differences are between the rows. For example, is it true that the bottom two rows used 563,030 annotations from QwQ-32B?  Then, what did each of the other rows use in terms of how many annotations they got and from what model they got them?  Can some intuition be given for what the cost is in terms of US Dollars or some intuitive measure of computing power or computing time or some appropriate measure(s) for readers that may not be deeply familiar with how much 563,030 annotations from QwQ-32B cost in terms of these more "practical" or more "commonly used" or more "intuitive" measures of cost?  Can similar measure(s) of cost be given for other rows so readers can more easily understand the amounts of the annotation cost savings of the various methods in more meaningful terms?

---

> ### Author Response · Authors · 2025-06-02
> **Response**
>
> Thank you for your thoughtful review. We sincerely appreciate your specific suggestions on symbol formatting and typographical corrections; these will be incorporated into the revised manuscript. Below are responses to the other **Weaknesses (W)** and **Questions (Q)**.
>
> ---
> **W1**: Some parts were not clearly written, particularly regarding interpretation of experimental results.
>
> We will revise the experiment results sections for greater clarity and concreteness. Detailed responses follow below.
>
> ---
>
> **Q1**: Do the bottom two rows in Table 1 both use 563,030 annotations from QwQ-32B?
>
> Yes. Both rows use the same 563,030 annotations from QwQ-32B. The distinction lies in their base models (lines 281-285).
>
> ---
>
> **Q2**: Can some intuition be given for what the cost is in terms of some intuitive measure, e.g., US dollars, for all methods?.
>
> The annotation costs (w.r.t. the number of generated tokens) for all methods are shown in Figure 3 and the estimation strategy are in Appendix C.
>
> For intuitive cost comparison, we estimate annotation costs in USD using OpenAI's pricing [1] (\$1.60 per 1M tokens). The following results are derived based on our results in Figure 3. Please note that models without open-sourced data (e.g. EurusPRM) couldn't be estimated.
>
> | Model                         | Est. Annotation Cost ($) |
> | ----------------------------- | ------------------------ |
> | ActPRM/ActPRM-X               | 1916                     |
> | Qwen2.5-Math-7B-Math-Shepherd | 30431                    |
> | Qwen2.5-Math-PRM-7B           | 33072                    |
> | Universal-PRM                 | 9168                     |
>
> ---
>
> **Q3**:  I think the ablation study on the number of heads for ensemble PRM would make sense to run with both epistemic and aleatoric uncertainty used for sampling.
>
> Thank you for your valuable suggestion. In response, we conducted an ablation study on the number of ensemble heads using both epistemic and aleatoric uncertainty for sampling (under $\delta_{\text{std}}=0.005$ and $\delta_{\text{pred}}=0.95$). The results below confirm that incorporating aleatoric uncertainty:
>
> Improves overall performance (F1 increases across head counts), and
> Reduces sensitivity to the number of heads (margins between configurations narrow).
>
> Nevertheless, 32 heads consistently achieves the highest F1 score (0.673), reinforcing its optimality for our approach.
>
> | Num. of Heads | Average F1 Score |
> | ------------- | ---------------- |
> | 2             | 0.658            |
> | 4             | 0.655            |
> | 8             | 0.663            |
> | 16            | 0.668            |
> | 32            | 0.673            |
> | 64            | 0.670            |
>
> ---
>
> **Q4**: In Figure 2a, why does the curve for the ActPRM model stop at Budget = approximately 0.6? Could it be run out all the way to Budget = 1.0?
>
> Figure 2a follows standard pool-based active learning [2][3], where models iteratively select samples from pre-labeled data. The curve stops at Budget=0.6 because ActPRM have assessed all data points but utilizing only 60% of the dataset - demonstrating 40% labeling cost savings. Extending to Budget=1.0 isn't applicable since:
>
> Budget=1.0 represents full-dataset training (our baseline), and
>
> ActPRM's core advantage is achieving comparable performance with reduced data requirements.
>
> ---
>
> **Q5**: It seems that the $\delta_{pred}$ and $\delta_{std}$ parameters were set to their best values via running through grid search over runs with a bunch of possible values. Does this give the proposed method an unfair advantage over other competing methods in the experimental results that are reported?
>
> We respectfully disagree. In controlled settings (Sec. 5.1), ActPRM consistently outperformed baselines. Figure 2a shows most hyperparameter configurations (semi-transparent points) surpass full-data tuning, demonstrating robustness. While we tuned hyperparameters following ML best practices, results indicate low sensitivity to these choices.
>
> ---
>
> **Q6**: Is it suggested that the values of 0.95 (confidence threshold) and 0.005 (standard deviation threshold) should now always be used with the proposed new method, even for new tasks and applications of the method beyond the datasets used in the paper's experiments?
>
> These values were optimal for our experiments and serve as starting points, not universal defaults. Hyperparameter tuning remains essential for new tasks/datasets—similar to tuning learning rates. Figure 2a suggests ActPRM’s performance is stable across nearby values, potentially easing adaptation..
>
> ---
>
> [1] https://openai.com/api/pricing
>
> [2] Gal, Yarin, Riashat Islam, and Zoubin Ghahramani. "Deep bayesian active learning with image data." International conference on machine learning. PMLR, 2017.
>
> [3] Dwaracherla V, Asghari S M, Hao B, et al. Efficient exploration for llms[J]. arXiv preprint arXiv:2402.00396, 2024.

---

> > ### Author Response · Authors · 2025-06-09
> >
> > Dear Reviewer p2Cc,
> >
> > Thank you for your time and review—we’d truly appreciate it if you could take a moment to review our rebuttal if you have a chance. Your insights are invaluable to us.
> >
> > Best regards,
> > Authors

---

### Decision · Program_Chairs · 2025-07-08

**Decision:**

Accept

**Comment:**

The paper proposes to reduce the cost of training process reward models by selecting examples to annotate based on the aleatoric and epistemic uncertainty, with the former characterized as P(step) and the latter as V[P(step)] across multiple prediction heads. This apparently makes it possible to attain better performance at significantly lower annotation cost (with annotation performed by a stronger reasoning model).

The reviewers are generally positive on the work, noting the high potential impact, strong motivation, and solid empirical results. Two minor potential concerns are (1) the tuning of the two hyperparameters (and the generality of the values chosen here), and (2) the limitation to settings in which a strong oracle annotator is available.